# Anti-Mitochondrial and Insecticidal Effects of Artemisinin against *Drosophila melanogaster*

**DOI:** 10.3390/ijms24086912

**Published:** 2023-04-07

**Authors:** Mengjiao Zhong, Chen Sun, Bing Zhou

**Affiliations:** 1School of Life Sciences, Tsinghua University, Beijing 100084, China; 2College of Agronomy and Biotechnology, China Agricultural University, Beijing 100193, China; 3Shenzhen Institute of Synthetic Biology, Shenzhen Institute of Advanced Technology, Chinese Academy of Sciences, Shenzhen 518055, China

**Keywords:** antifeedant, membrane potential, mitochondrion, insects

## Abstract

Artemisinin (ART) is an endoperoxide molecule derived from the medicinal plant *Artemisia annua* L. and is clinically used as an antimalarial drug. As a secondary metabolite, the benefit of ART production to the host plant and the possible associated mechanism are not understood. It has previously been reported that *Artemisia annua* L. extract or ART can inhibit both insect feeding behaviors and growth; however, it is not known whether these effects are independent of each other, i.e., if growth inhibition is a direct outcome of the drug’s antifeeding activity. Using the lab model organism *Drosophila melanogaster*, we demonstrated that ART repels the feeding of larvae. Nevertheless, feeding inhibition was insufficient to explain its toxicity on fly larval growth. We revealed that ART provoked a strong and instant depolarization when applied to isolated mitochondria from *Drosophila* while exerting little effect on mitochondria isolated from mice tissues. Thus, ART benefits its host plant through two distinct activities on the insect: a feeding-repelling action and a potent anti-mitochondrial action which may underlie its insect inhibitory activities.

## 1. Introduction

Artemisinin (ART), an important natural product derived from the plant *Artemisia annua* L., has recently gained much attention due to its outstanding antimalarial properties, especially in light of the emerging drug resistance seen with other antimalarial drugs. Awarding the 2015 Nobel Prize in Physiology or Medicine for the discovery of ART also helped to spotlight this antimalarial drug. 

ART is a unique natural product containing a pharmacologically relevant endoperoxide bridge (Figure 1A) situated in the backbone structure of a sesquiterpene lactone [1,2,3]. In addition to its prominent effects as an antimalarial drug, ART also exhibits activities against other parasites such as *Toxoplasma* [4], *Leishmania* [5], *Clonorchis* [6], *Schistosoma* [7], and even some cancer cells [8,9] and viruses [10,11], albeit with a lower efficacy.

In the course of studying the effects of ART on various lab organisms, we initially intended to use *Drosophila melanogaster* as a negative control but unexpectedly found that ART conferred inhibitory effects on this organism. A literature search revealed that there have been quite a few reports about the inhibitory activities of *A. annua* plant extracts, including ART, on various insects [12,13,14,15,16,17,18,19]. From a botanical perspective, ART production may confer some advantages to the host plant. Agricultural reports and previous research suggest that *A. annua* might be toxic to the soil and aquatic organisms, inhibit the growth of other plant species, and might also exhibit insecticidal, antifungal, and antibacterial effects [12,13,14,20,21,22,23]. Regarding its anti-insect effects, it has been previously shown that *A. annua* extracts, or even the ART compound itself, can reduce insect feeding and inhibit their growth. However, to our knowledge, no concrete evidence exists to show that the insecticidal effect of ART is solely due to the antifeedant activity of ART, either alone or in combination with other activities. *D. melanogaster*, a popular lab model organism, has been previously used in the evaluation of insecticidal effects of many natural compounds such as naringenin and ingredients from basidiomycete mushrooms or mistletoe [24,25,26]. In this work, we used *D. melanogaster* to show that ART possesses two distinct anti-insect activities, a moderate antifeeding activity, and an insecticidal activity. Importantly, we detected that ART has a strong direct effect on fly mitochondria. Our data suggest ART might protect the host plants from insect infestation through mitochondrial interference.

## 2. Results

### 2.1. ART Negatively Affects D. melanogaster in a Concentration-Dependent Manner

We evaluated in detail the potential effects of ART on the fly. As shown in Figure 1B, a developmental delay of larvae raised on ART food was observed. After three days of incubation at 25 °C, almost all the larvae on the regular food (normal food) grew to the 3rd instar stage, while the larvae on the ART food grew only to the size of a typical 2nd instar larva. Compared with the control, there was a dark yellowish-brown area deep below the cuticle of the 3rd instar larvae which had consumed ART-laced food (Figure 1C). Larval dissections revealed that parts of the midgut were affected (Figure 1D), although the underlying reason for the color change was unclear. By the sixth day, the larvae on the regular food developed into pupa, while those on the ART food remained as wandering larvae at various development stages (Figure 1E). The ART effect on eclosion was also measured. At 25 °C, low (50 μM) or median (100 μM, 200 μM) levels of ART feeding did not notably affect the eclosion rate as compared with the normal food (NF) control (0 μM); however, high concentrations (400 μM) of ART led to a significantly reduced pupal formation and a drop in the eclosion rate (Figure 1F). At a concentration of 400 μM ART, some larvae died and turned black. In summary, at 25 °C, low levels of ART significantly retard larval growth without being lethal, while higher levels are fatal to a small larvae population.

We then set out to ascertain whether ART could have an even longer-term effect, i.e., whether continued negative influence could be observed in adult flies. The movement and lifespan of files newly enclosed from normal food were recorded. As seen in Figure 1G–J, ART shortens the lifespan and affects the movement activity of *Drosophila*, with female flies showing more sensitivity to ART than male flies. For the male flies, no obvious drop in locomotive abilities was seen at feeding concentrations of 200 μM or less of ART, but with the higher concentration (400 μM) of ART, a sharp decrease in climbing was observed (Figure 1G). For females, impairment of locomotive ability was observed at concentrations as low as 100 μM ART, which progressively worsened at higher levels of ART (Figure 1I). A similar trend was noticed in the longevity assay. Median concentrations (100 μM, 200 μM) of ART led to dramatically reduced lifespans of female flies, with a median lifespan of only 43 days at a concentration of 400 μM ART, compared with over 70 days at low (50 μM) or no ART (Figure 1J). For the male adult flies, only the higher concentration of ART (400 μM) showed a sharp decrease in longevity (Figure 1H).

### 2.2. Elevated Temperatures Exacerbate the Action of ART

At 29 °C, fruit flies develop at about twice the rate seen at 25 °C and without any observed abnormalities. Therefore, 29 °C is a temperature often adopted to speed up growth in the laboratory. Because the previous experiments had been performed at 25 °C, we wanted to rerun them at a higher temperature. Amazingly, with a mere increase of 4 degrees, ART conferred a dramatically more severe effect on the fruit fly. A concentration of 400 μM ART was almost entirely lethal (Figure 2A), compared with the much lower toxicity observed when a concentration of 400 μM ART was delivered at 25 °C. In addition, noticeable effects on eclosion were observed starting at concentrations as low as 100 μM ART at 29 °C. In comparison, at 25 °C these effects were hardly noticeable at the higher concentration of 200 μM ART (Figure 1F).

One point to consider is whether or not the drug concentrations used in these experiments are abnormally high. For example, malaria parasites can be inhibited by nM concentrations of ART, whereas in our experiments, we administered μM concentrations of ART. Are similar effects achievable with regular plant leaves? To verify in a more physiologically relevant situation the toxicity effects of ART produced by the plant, we collected *A. annua* leaf samples from different regions of China. *A. annua* widely grows in south China, including China’s Sichuan, Hunan, and Guangxi provinces. It is known that ART amounts could vary depending on the batches. The collected leaves were pulverized, and the superfine powders were thoroughly mixed with the standard cornmeal medium. Similar to the above-mentioned effect of ART, our results clearly indicate that the administration of leaf powders also sharply reduced the eclosion rate in a concentration-dependent manner (Figure 2B).

### 2.3. ART Possesses Distinctly Separable Antifeeding and Insecticidal Activities

One question is whether the inhibitory effect of ART on the fly is due to its toxicity or its repelling effect. Conceivably, the drug could be so distasteful that the insect might choose hunger rather than feeding on it. The observed phenotypes of developmental delay and decreased eclosion rate may thus arise from malnutrition rather than the toxicity of the drug per se. In order to test these possibilities, we first analyzed whether the larvae indeed consumed the food. A red food dye was used to indicate food intake. As shown in Figure 3A, a distinct red color was observed for all the larvae fed with ART and two control compounds, quinine (QUI), a bitter-tasting chemical, and its sulfate form quinine sulfate dihydrate (QUI-S). The intensity of the redness in each larva roughly correlated with the amount of food intake. The larvae were able to feed ad libitum, although it appeared that ART and QUI were consumed less. To further quantify the feeding avoidance behavior, we designed a taste and feeding tendency test for ART and two other natural compounds, QUI and curcumin (CUM), which have also been clinically used. We allowed the larvae to choose between pure agarose (PURE) and agarose with different compounds added. As shown in Figure 3B, prominent avoidance of QUI/QUI-S was observed. ART also seemed to taste bad but was not avoided as much as QUI. The repelling activity became more evident at higher concentrations for both ART and QUI. Interestingly, flies seemed to favor CUM to some extent.

Under these compounds concentrations, we measured flies’ eclosion rates to test if the levels of preference for the food might result in the differences observed in the flies’ development. The results shown in Figure 3C indicate that although flies do not prefer a drug-laced meal, the repelling taste of ART could not fully explain its inhibitory effect. Although the flies disliked quinine more than ART, the inhibition of development and eclosion was much more pronounced on the ART food, suggesting ART is additionally toxic to the fly separate from its repelling effect. Worth noting is that the repelling effect of *A. annua* leaf powders was similar to that of pure ART, implying ART by itself could explain the repelling effect of *A. annua* leaves.

### 2.4. Drosophila Mitochondria Are Sensitive to the Direct Action of ART

ART has been shown to be able to instantly and directly depolarize malarial and yeast mitochondria [27,28,29,30]. Based on the above observation that the movement of *Drosophila* is sensitive to ART it is natural to test a connection between ART and mitochondria. In *Drosophila*, when mitochondria function is compromised, the first tissues affected are the muscle and nervous system because they are the most energy-demanding regions. We examined the mitochondrial membrane potential of fruit flies treated with ART. The uptake of Rh123 was used to indicate the membrane potential changes. As shown in Figure 4, the membrane potential of ART-treated *Drosophila* mitochondria was much lower than that of the control mitochondria. For the control, we isolated mitochondria from the mouse brain and tested its membrane potential under ART treatment with the same experimental conditions. As shown in Figure 4, even when incubated with 100 μM ART, 10-fold higher than that used in the *Drosophila* reaction system, the membrane potential of mouse mitochondria was only slightly affected. Taken together, these experimental results demonstrate that ART directly and selectively causes mitochondria dysfunction in the fly and suggest this activity could be one of the important reasons, if not the sole one, underlying the selective insecticidal activity of *A. annua*. 

## 3. Discussion

The discovery of ART nearly half a century ago has had a great significance and positive influence on human health. On the agricultural side, it is known that *A. annua* exhibits superior growth over the other plants grown in close proximity [20,31], and it is known that farmers in China mix *A. annua* leaves or its extractions into the water to make a natural herbicide or pesticide. Research has also been published about the biological activities of *A. annua* against microorganisms [32,33], insects, and certain other invertebrates [16,20,34], and its phytotoxic properties to crops [35,36], weeds [35], aquatic plants [37,38], and even to lettuce and radishes [23,31,37]. Most of these studies used extracts of *A. annua* and only occasionally ARTs. In these previous studies, antifeeding and growth inhibition were also observed. However, it was not known whether these two activities are distinct or separable. For example, two previous studies observed the feeding-deterring properties of *A. annua* extracts and ART against *Epilachnapaenulata*, *Spodopteraeridania*, and *Cydiapomonella* [12,15]. A possible neurotoxic effect of ART against *E. paenulata* was also indicated [15]. However, the authors were not able to attribute this to feeding inhibition or neurotoxicity per se. In the present study, we have demonstrated that pure ART elicits insecticidal effects in addition to its antifeedant activity. 

Previous field studies have shown that ART is mainly concentrated in the leaves of the plant (more precisely, it mainly accumulates in the glandular trichomes present on the surface of the leaves), although it has also been observed in the corolla and receptacle of the florets [20,39,40,41]. Variable contents of ART in the dry leaves of the plant have been reported in the literature, ranging between 0.01 to 1.0% depending on the source [42,43,44]. Interestingly, a big part of the natural range of ART content approximately corresponds to the concentrations we used in this study, or even higher, suggesting the concentrations we used are physiologically relevant. In other words, the levels of ART in the host plant might be “designed” in such a way that it is just enough to inhibit insects. Although at moderate concentrations, ART is not lethal to the larvae in the lab condition, it is worthwhile to point out that a slight compromise in fitness may be detrimental in the wild where competition is high and surviving conditions could be much harsher.

To our surprise, a slight elevation of temperature could obviously exacerbate ART toxicity. At 29 °C, the toxic effect of ART is significantly more potent than that at 25 °C. Mitochondrial respiration enhancement might be one possible explanation, but more research is required to make a definite conclusion. Worth noting is that unlike laboratory conditions, where the temperature is usually kept at 25 °C to rear up flies, temperatures of 29 °C or higher are typically seen in many parts of south China in the summer, where *A. annua* thrives and is abundant. We, therefore, speculate that ART may be more of an insecticide in tropical or subtropical regions where the temperature is higher.

From earlier work, our group and other research groups have demonstrated that ART can quickly exert depolarization actions on both malaria parasites and yeast mitochondria [28,29,45,46], accompanied by ROS production, but without respiration inhibition. However, the action of ARTs (ART and its derivatives) is not limited to their anti-mitochondrial mode. In anti-cancer activities, ARTs may work through another type of action, specifically heme-mediated activation [27,30,47]. Interestingly, in this work, we found that ART most likely works against *Drosophila* in a mode somewhat similar to its action against yeast and malarial parasites. It is not clear what common factor of these mitochondria confers sensitivity to ART, which remains one of the most important questions to be answered.

## 4. Materials and Methods

### 4.1. Fly Stocks, Culture Media Preparation

*Drosophila w^1118^* was used in this study. Flies were reared on standard cornmeal media (normal food, NF) with a 12 h light/dark cycle at 25 °C unless otherwise indicated (29 °C). Pure ART was purchased from Chengdu Okay Medicine Co., Ltd. (Chengdu, China). ART was prepared in DMSO as 200 mM stock and stored at −20 °C for later use. To prepare diets with the drug, ART stock was diluted with DMSO to proper concentrations and thoroughly mixed with the fly food to ensure even distribution. NF groups were also added with an equal volume of DMSO as the controls. *A. annua* leaves were pulverized, and the superfine powder was directly added and completely mixed into the standard cornmeal medium.

### 4.2. Eclosion Assay

To examine the effects of ART on *Drosophila* development, flies were put on grape juice agar plates at 25 °C to lay eggs. First instar larvae were collected and then transferred to 10 cm (L) × 2.5 cm (D) glass vials containing 8 mL NF (control) or ART-supplemented media at a density of 30 larvae per vial kept at 25 °C or 29 °C. The total number of pupae and emerging adults in each vial was counted. Each concentration of the experimental groups was tested in quadruplicate.

*A. annua* leaves were purchased from different resources, frozen in liquid nitrogen, and ground into fine powders. To test the effect of the *A. annua* leaf powers on *Drosophila* development, 1st instar larvae were transferred to glass vials containing media with NF, 20%, 40%, or 60% (*m*/*v*) of these powders at the density of 30 larvae per vial. These larvae were then grown at 29 °C. The number of enclosed flies was recorded.

### 4.3. Longevity Assay

Newly enclosed flies from normal food were used to carry out this assay. Flies were separated by sex and transferred to a fresh culture medium. Twenty flies were placed in each vial and all vials were kept at 25 °C or 29 °C with 60% humidity. Vials were changed every two days. All of the flies were transferred without anesthetization and the mortality, if any, was recorded. The experiments continued until about day 90 from the beginning, if applicable. Eight parallel groups were conducted for each feeding regime.

### 4.4. Climbing Assay

To measure mobility, adult flies were fed with diets containing different concentrations of ART, transferred into the climbing ability test vials, and incubated for 1 h at 25 °C to allow for environmental acclimation. After tapping the flies down to the bottom of the vial, the number of flies that climbed over a marker line at 7 cm on the vial within 8 s was counted. During the climbing test, the flies were assessed in consecutive trials separated by 1 min of rest. Four trials and five replicates for each trial were observed and recorded.

### 4.5. Taste and Feeding Tendency Assay

To measure feeding behavior towards drugs, we followed a procedure based on [48]: 10 larvae were placed on a Petri dish filled with 1% agarose plus 0.2% red food dye (amaranth) with or without the chosen concentration of drugs. The larvae were allowed to feed on either of these respective substrates for 15 min; they were then washed in water and placed on ice for approximately 2 min. Finally, larvae were put into boiling water for 2 s and transferred to a new dish for photography.

To more quantitatively analyze feeding preferences, we performed the experiment as previously described [48] with some modifications (Figure 5). Twelve hours before experiments, 60 mm Petri dishes (CORNING 430166, Corning Incorporated, Corning, NY, USA) were prepared by first creating a 0.8 cm wide “buffer” zone down the middle of the plate. Onto one side of the “buffer” zone, the plate was filled with only 1% agarose (henceforth called PURE) and the other half with 1% agarose plus different drugs, including ART, quinine (QUI), quinine sulfate dihydrate (QUI-S), and curcumin (CUM) at the respectively indicated concentrations. For the control condition, both sides and the “buffer” zone of the Petri dish contain only pure agarose. For each test, 15 larvae were placed in the middle of the dish (the “buffer” zone). We recorded the number of larvae on either side of the dish and calculated a taste and feeding preference index (TFPI) as: TFPI=(#DRUG−#PURE)/#TOTAL. In this equation, # indicates the number of larvae on the respective side of the dish. Thus, the variation of the index values will range between −1 to 1, indicating avoidance to preference for drugs. Scoring was taken 15 min after the larvae were put onto the dishes, and experiments were repeated at least 15 times for each experiment.

### 4.6. Mitochondria Isolation

*Drosophila* and mice mitochondria were isolated according to the procedure as published [49], with slight modifications. Briefly, about 200 flies or one mouse brain (gifted by the Experimental Animal Center of Tsinghua University) per isolation were collected and placed on ice. They were then homogenized in a total of 10 mL of extraction buffer (250 mM sucrose, 5 mM Tris–HCl, 2 mM EGTA, 1% (*w*/*v*) bovine serum albumin, pH 7.4 at 4 °C) with a 15 mL conical glass tissue homogenizer. We found that better extraction efficiency could be achieved using about 30 flies in 1.5 mL buffer for each homogenization. The homogenate was centrifuged at 1000× *g*, 4 °C, for 10 min. The supernatant was transferred to a new, cold 30 mL conical tube and was re-centrifuged at 4 °C, 1000× *g* for 10 min. This step was repeated until there were no distinct particles in the precipitation. The supernatant was then pelleted to yield crude mitochondria fraction by centrifugation at 4 °C, 10,000× *g* for 10 min. For further purification, a sucrose gradient ultracentrifugation was performed for 1 h at 134,000× *g*, 4 °C, yielding highly pure mitochondria. The protein concentration of the mitochondria was estimated by a standard BCA kit (Thermo Fisher, Waltham, MA, USA).

### 4.7. Determination of Mitochondrial Membrane Potential

Membrane potentials (Δψ) of the isolated *Drosophila* and mice mitochondria were assessed by measuring the Δψ-dependent uptake of rhodamine 123 (Rh123). Isolated mitochondria (1 μg/μL protein) were incubated at 25 °C in a buffer containing 120 mM KCl, 5 mM KH_2_PO_4_, 3 mM HEPES, 1 mM EGTA, 1 mM MgCl_2_, 0.2% BSA, and pH 7.2. Δψ was assessed by measuring the uptake of Rh123 with a microplate fluorometer (Fluoroskan Ascent, Thermo, Waltham, MA, USA) at 485 nm excitation and 538 nm emission after the addition of Rh123 to the mitochondria suspension at 50 nM final concentration.

### 4.8. Statistical Analysis

All data are presented as mean ± standard error of the mean (SEM) for at least three independent experiments. Statistical analyses were carried out by one-way ANOVA followed by Tukey’s test in multiple groups. Unpaired two-tailed *t*-tests were performed in two groups. Statistical details are indicated in the figure legends.

## Figures and Tables

**Figure 1 ijms-24-06912-f001:**
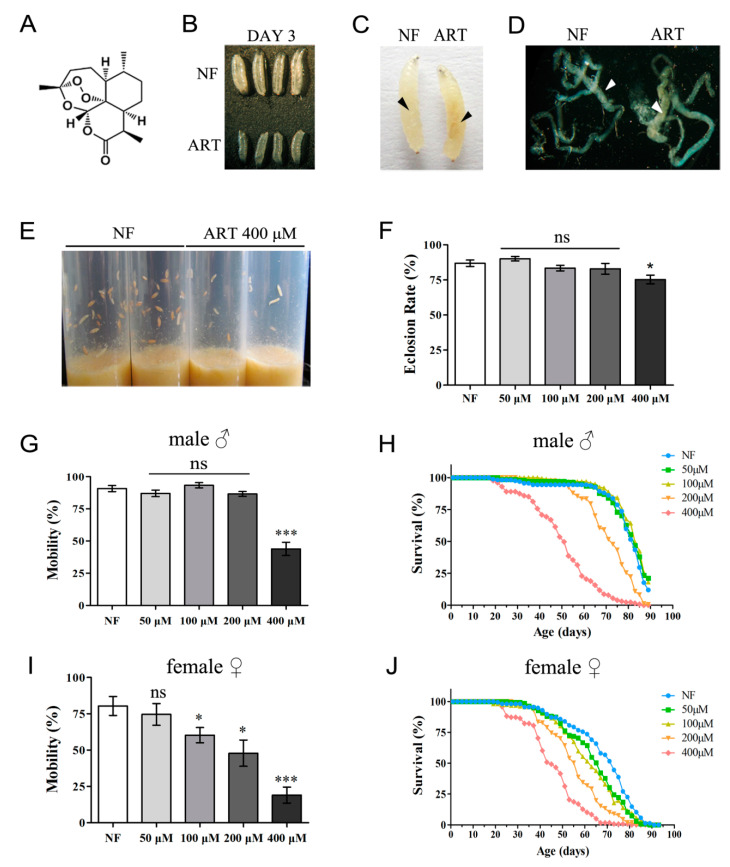
ART negatively impacted *Drosophila* in a concentration-dependent manner at 25 °C. (**A**) The molecular structure of ART. Note a unique endoperoxide bond situated in the sesquiterpene backbone. (**B**) Effect of ART on the growth and development of *Drosophila* larvae. After three days of culturing, typical growth of the larvae fed on normal food (NF) or 400 μM ART food was shown. (**C**,**D**) When the larvae reach a similar size (which takes longer for larvae grown on ART food), the larvae fed with ART food have a dark yellow midgut (black arrows in C and white arrowheads in D indicate the changed areas). (**E**) An obvious developmental delay on ART food. Food was provided with the standard cornmeal as the NF and mixed with 400 μM ART as the ART food. (**F**) The eclosion rates of the fly on food mixed with a set of concentrations of ART were measured, and a significant decrease in eclosion rate could be observed on high levels of drug diet. (**G**) The mobility defect could only be seen at high concentrations of ART food (400 μM) for male flies. (**I**) Dosage-dependent mobility defect of female flies was observed. (**H**,**J**) ART shortened the lifespan of male flies and female flies. The *x*-axis refers to the age of adult flies. The longevity changes on the ART diet are correlated with the phenotype of movement disorder for both female and male flies. Flies were reared at 25 °C in this experiment. NF, normal food. Values are presented as mean ± SEM; *n* ≥ 5. *** *p* < 0.001, * *p* < 0.05. ns, no significance.

**Figure 2 ijms-24-06912-f002:**
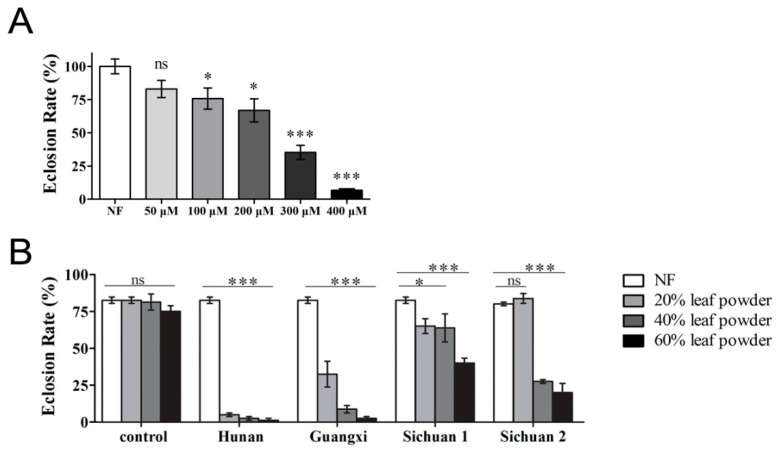
Elevated temperatures exacerbated the effect of ART on the flies. (**A**) A dramatic decrease in eclosion rate at 29 °C, a mere increase of 4 degrees, was observed when flies were treated with ART. (**B**) Powdered leaves of *A. annua* from different regions exhibited similar, albeit various, levels of inhibitory effects on flies. Food was prepared from the standard corn meal mixed with the leaf powders. The eclosion rate was assayed at 29 °C. Data are presented as mean ± SEM; *n* ≥ 5. *** *p* < 0.001, * *p* < 0.05. ns, no significance.

**Figure 3 ijms-24-06912-f003:**
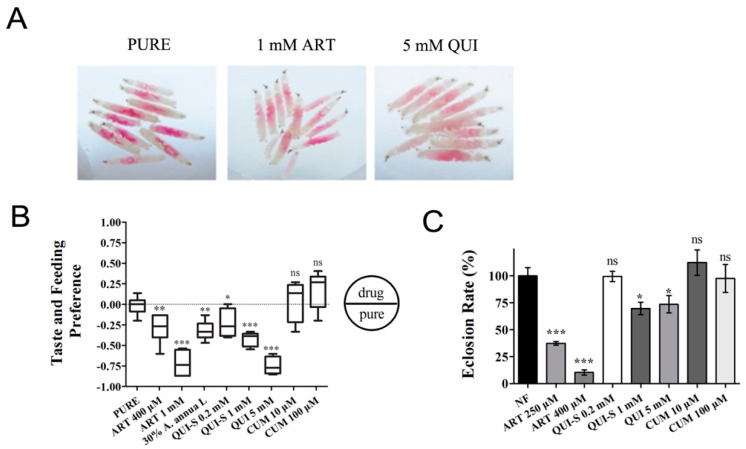
ART is a weak antifeedant but possesses a distinct insecticidal activity. The feeding tendency of *Drosophila* larvae on different natural compounds is shown. (**A**) Larvae feeding assay. Larvae are allowed to feed on either a red−dyed plate or a red−dyed plate with ART or QUI, for 15 min. Larvae ate less in ART− or QUI−added food, as indicated by the intensity of the dye in the gut. *n* ≥ 3. (**B**) Larvae feeding tendency assay. Early 3rd instar larvae were allowed to freely choose between the two sides of a petri dish containing 1% agarose with or without the drugs: artemisinin (ART), quinine (QUI), quinine sulfate dihydrate (QUI−S), and curcumin (CUM). DMSO was used as a control (PURE). A feeding preference index was calculated to indicate the tendency of larvae toward the compound. *n* ≥ 10. (**C**) The eclosion rate of flies on different drugs. These data show that under these concentrations, ART is a weaker antifeedant but a much stronger insecticide. The files were kept at 29 °C. *** *p* < 0.001, ** *p* < 0.01, * *p* < 0.05. ns, no significance.

**Figure 4 ijms-24-06912-f004:**
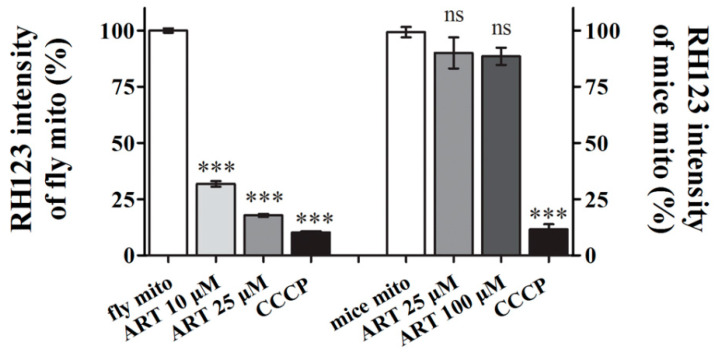
ART is a potent *Drosophila* mitochondrial depolarizer. ART selectively depolarizes *Drosophila* mitochondria but not mice mitochondria. Mitochondria isolated from *Drosophila* and mouse brains were respectively treated with two concentrations of ART. Δψ (mitochondrial membrane potential) was monitored by the fluorescence intensity of rhodamine 123 (Rh123). A higher fluorescence intensity of stained cells indicates a higher Δψ. CCCP (Carbonyl cyanide 3-chlorophenylhydrazone) was used as the positive depolarizer control. ART treatment resulted in a dramatic loss of Δψ in *Drosophila* mitochondria but little effect on the mammalian mitochondria even at a much higher concentration. Data are shown as mean ± SEM. *n* ≥ 5. *** *p* < 0.001. ns: no significance.

**Figure 5 ijms-24-06912-f005:**
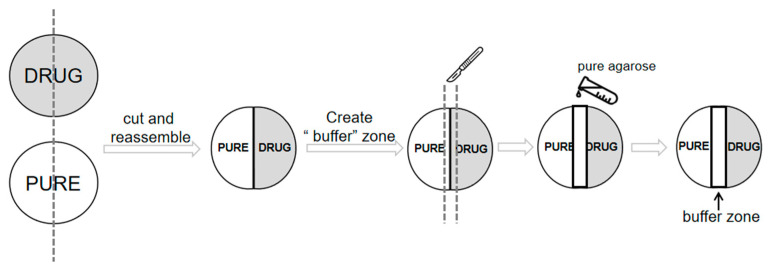
Creating a feeding assay plate. The same volume of prepared agarose solutions with or without the drug was poured into different 60 mm Petri dishes. The drug plate (DRUG) and PURE plate (no drug) were divided into two pieces and recombined. In order to better bridge the two parts of the new plate and establish an area for the larvae to freely choose, a 0.8 cm-wide middle zone was created in the middle of the plate with a sharp knife and refilled with pure agarose. All larvae to be tested were initially placed in the middle area before taste and feeding tendency assays started.

## Data Availability

All data are included in the manuscript.

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
