# Peer review of "Anti-Mitochondrial and Insecticidal Effects of Artemisinin against Drosophila melanogaster"

_ijms, 2023, doi:10.3390/ijms24086912_

Round 1

Reviewer 1 Report

This is an interesting piece of work. How artemisinin, a metabolite of A. annual and an antimalarial drug, benefits the host plant is not clarified. This research investigated the anti-insect effect of artemisinin and helped us understand the native function of this important chemcial, i.e., how A. annual uses this to defend against insects (in two distinct ways). Before its acceptance, some minor revisions are recommended.

1-Page 2, paragraph 3, "D. melanogaster" should be "Drosophila melanogaster" for the first time the organism's name was used in the main text. In the same paragraph, line 10, "Drosophila melanogaster" should be "D. melanogaster" or "Drosophila".

2-In the method section, the statistical analysis and method should be included.

3-When the concentration was below 100 uM, the effect of Art on male and female flies was different. Any reason to explain the difference? Also, any speculation for the different responses of flies' mitochondria and mice's mitochondria to ART? 

4-In Figure 2B and Figure 3B, statistical analyses and their significance should be indicated in the figures. Similarly, in Figure 4, the effect and significance of CCCP should also be analyzed and included.

5-Figure 3A and the legend, drug concentration should be indicated.

Reviewer 2 Report

Zhong et al

Artemisinin is anti-mitochondrial and insecticidal against Drosophila melanogaster

This is a nice paper, well written on the basis of well executed experimentation. Some additional information is needed, however.

Here go my recommendations:

Lines 74 and 84: what were illumination conditions? Total darkness or some photo regime, maybe?

Lines 82-88: Please make it clear that either synthetic ART or plant powders were used

Lines 89-95: how ART was applied here? I assume that adults eclosed from larvae treated with ART in eclosion assay were used, but clear explanation is needed

Lines 96-103: Did the authors weigh the flies? Maybe differences in climbing ability come from differences in body mass?

Lines 104-110: What was the aim of this experiment? The authors photographed the larvae, but did they analyze the pictures, for instance for optical density or percentage of stained larvae?

Lines 111-126: More explanation id needed here. The authors indicate the work of El-Keredy (2012) as an inspiration for their bioassay, but their methodology differs markedly from the original paper. In particular, it is not clear how “buffer zone” was created. Was it a recession in the agarose layer? Was the test arena created by placing three different pieces of agar? A photography or a drawing of test arena would help a lot.

Lines 141-148 and 260-262: The authors use mice mitochondria for comparison (lines 260-262) but do not mention that in Materials and Methods (lines 141-148). Please add that information.

Line 147: Exactly what was the concentration of Rh123 in the suspension?

Materials and Methods: What statistics was used for evaluation of data?

Line s 182-204: There is no mention about this experiment in Materials and Methods. Please provide that information.

My recommendation is publishing the manuscript after minor corrections.
